# picoSMMS: Development and Validation of a Low-Cost and Open-Source Soil Moisture Monitoring Station

**DOI:** 10.3390/s25226907

**Published:** 2025-11-12

**Authors:** Veethahavya Kootanoor Sheshadrivasan, Jakub Langhammer, Lena Scheiffele, Jakob Terschlüsen, Till Francke

**Affiliations:** 1Department of Physical Geography and Geoecology, Faculty of Science, Charles University, 128 00 Prague, Czech Republic; jakub.langhammer@natur.cuni.cz; 2Institute of Environmental Science and Geography, University of Potsdam, 14476 Potsdam, Germany; lena.scheiffele@uni-potsdam.de (L.S.); terschluesen@uni-potsdam.de (J.T.); francke@uni-potsdam.de (T.F.)

**Keywords:** soil moisture sensor, capacitive sensor, low-cost sensor, open source, monitoring station

## Abstract

Soil moisture exhibits high spatio-temporal variability that necessitates dense monitoring networks, yet the cost of commercial sensors often limits widespread deployment. Despite the mass production of low-cost capacitive soil moisture sensors driven by IoT applications, significant gaps remain in their robust characterisation and in the availability of open-source, reproducible monitoring systems. This study pursues two primary objectives: (1) to develop an open-source, low-cost, off-grid soil moisture monitoring station (picoSMMS) and (2) to conduct a sensor-unit-specific calibration of a popular low-cost capacitive soil moisture sensor (LCSMS; DFRobot SEN0193) by relating its raw output to bulk static relative dielectric permittivity ($\epsilon_{s}$), with the additional aim of transferring technological gains from consumer electronics to hydrological monitoring while fostering community-driven improvements. The picoSMMS was built using readily available consumer electronics and programmed in MicroPython. Laboratory calibration followed standardised protocols using reference media spanning permittivities from 1.0 (air) to approximately 80.0 (water) under non-conducting, non-relaxing conditions at 25 ± 1 °C with temperature-dependency characterisation. Models were developed relating the sensor’s output and temperature to $\epsilon_{s}$. Within the target permittivity range (2.5–35.5), the LCSMS achieved a mean absolute error of 1.29 ± 1.07, corresponding to an absolute error of 0.02 ± 0.01 in volumetric water content (VWC). Benchmarking revealed that the LCSMS is competitive with the ML2 ThetaProbe, and outperforms the PR2/6 ProfileProbe, but is less accurate than the SMT100. Notably, applying the air–water normalisation procedure to benchmark sensors significantly improved their performance, particularly for the ML2 ThetaProbe and PR2/6 ProfileProbe. A brief field deployment demonstrated the picoSMMS’s ability to closely track co-located HydraProbe sensors. Important limitations include the following: inter-sensor variability assessment was limited by the small sensor ensemble (only two units), and with a larger sample size, the LCSMS may exhibit greater variability, potentially resulting in larger prediction errors; the characterisation was conducted under non-saline conditions and may not apply to peat or high-clay soils; the calibration is best suited for the target permittivity range (2.5–35.5) typical of mineral soils; and the brief field deployment was insufficient for long-term validation. Future work should assess inter-sensor variability across larger sensor populations, characterise the LCSMS under varying salinity, and conduct long-term field validation.

## 1. Introduction

Soil moisture, also referred to as soil water content (SWC), is a pivotal geophysical state variable that holds importance across diverse fields such as hydrology, meteorology, agriculture, and ecology, among others. In the context of hydrology, SWC significantly influences vital hydrological fluxes, playing a significant role in partitioning the incoming precipitation and outgoing evapotranspiration to and from the terrestrial water storages respectively, thus influencing the rainfall-runoff response, especially where saturation-excess runoff processes dominate, and profoundly impacts phenomena like drought, floods, and groundwater recharge [1,2,3,4,5]. Understanding these interactions is essential for informed decision-making in water resource management, especially in the face of changing hydrological regimes arising from climate- and land-use- changes. Beyond its hydrological significance, SWC is fundamental to agronomy, dictating crop water needs and affecting overall crop growth, and is also vital for the ecological and biological characteristics of soil. It influences the mechanical properties of soil, such as consistency, compatibility, cracking, swelling, shrinkage, and density, thus affecting slope-stability in vulnerable areas [6].

The quantification of SWC has necessitated the development of various measurement techniques. The historically used thermogravimetric method is still used as a direct and reference method for SWC measurement, based on weight differences before and after drying of a controlled volume of bulk soil. While highly accurate, it is destructive, laborious, time-consuming, and unsuitable for real-time or large-scale field studies, especially for applications requiring frequent, in situ observations. Other early techniques, such as the neutron moderation technique, while in regular use, present limitations due to radiation hazards and the need for site-specific calibration. This has driven the development of indirect, non-destructive methods, including neutron scattering, dielectric techniques (Time Domain Reflectometry (TDR), Transmission Line Oscillator (TLO), Frequency Domain Reflectometry (FDR), and capacitance probes), electrical resistivity, heat pulse, optical techniques, gamma attenuation, Cosmic-ray Neutron Sensing, and Micro Electro Mechanical Systems (MEMS) [1,2,3].

SWC exhibits significant spatial and temporal variability, influenced by atmospheric conditions, topography, soil properties, and vegetation; a deeper understanding of these variations is crucial for comprehending hydrological processes, improving rainfall-runoff response, and enhancing ecohydrological insights [4,5]. To capture this inherent variability, extensive monitoring efforts are necessary, requiring dense networks of soil moisture sensors (SMSs), to quantify lateral variation, over large areas with high spatial and temporal resolution [4,5,7,8,9]. However, traditional SMSs, while tested robustly, are often expensive, which limits their widespread deployment for such comprehensive monitoring. This is where low-cost SMSs, despite their lower accuracy, become a viable option. Their affordability enables the acquisition and deployment of a greater number of sensor nodes, allowing for more detailed spatial and temporal observations that would otherwise be constrained by budget limitations [7,8,10,11,12]. By providing this enhanced data density, low-cost sensor networks significantly contribute to a better understanding of hydrosystems.

The mass production of capacitive SMS, driven by their widespread integration into *smart-things* and *Internet of Things (IoT)* applications in agriculture and gardening-related consumer electronic products, has made them highly affordable [13,14,15], with some models priced as low as USD 2–USD 10 [16,17]. While commercial sensors offer higher accuracy and reliability at a significant acquisition cost—often ranging from USD 150 to USD 5000 [10,16,17] for professional systems (sensors alone)—low-cost sensors, despite typically offering acceptable accuracy, enable larger deployments with a lower investment. This low cost is particularly advantageous in facilitating the maximisation of spatio-temporal extension in monitoring networks. These capacitive sensors operate on the fundamental principle that water possesses a significantly higher bulk static relative dielectric permittivity (≈80) compared to dry soil (≈5) and air (≈1). Consequently, the bulk static relative dielectric permittivity (hereafter referred to as $\epsilon_{s}$ or simply permittivity) of the soil is predominantly influenced by its water content [3,18]. These sensors employ a capacitor, often etched onto the printed circuit board (PCB) as two distinct conductive traces that are embedded within the soil matrix which serves as the dielectric medium. An R-C circuit measures the capacitor’s charging time and reports it as an analog voltage, a value inversely proportional to the soil’s permittivity and, consequently, inversely proportional to the SWC. In the interest of avoiding redundancy, readers who wish to inform themselves deeper on the measuring principle are redirected to the following studies: Susha Lekshmi S.U. et al. [2], Terzic et al. [18], Tarantino et al. [19], Placidi et al. [20], and Mane et al. [3].

The widespread availability of these mass-produced sensors, such as the popular model by DFROBOT bearing the stock keeping unit (SKU) “SEN0193” (hereafter referred to as LCSMS), presents a significant opportunity for hydrological research. This particular sensor was chosen for its robust build quality and minimal sensor-to-sensor variability compared to similar unmarked sensors, often sold under generic titles like “Capacitive Soil Moisture Sensor v1.0, v1.2, or v2.0” [16,21,22,23]. The sensor’s design, which incorporates a TLC555 timer integrated circuit (IC) operating in astable mode, generates pulses at 1.5 MHz [16,17]. This high frequency has been shown to reduce fringe-capacitance effects, effectively making it a contact-measuring device.

A considerable body of research has focused on evaluating the performance of LCSMS. However, the reliability of some studies is questionable. While many claim to evaluate the DFRobot SEN0193, accompanying images often suggest that a different, visually similar sensor was used such as in Schwamback et al. [10], Majumder et al. [13], Abdelmoneim et al. [14], Placidi et al. [16,20], Nagahage et al. [21], Kulmány et al. [24], Chowdhury et al. [25]. This is a critical issue, as variations in design and components can lead to significant sensor-to-sensor variability [14,21], potentially rendering the findings of such studies unreliable for the more consistent LCSMS.

Despite these inconsistencies, a wide array of efforts have been made to test and validate what is reported as the LCSMS sensor. The most common approach involves laboratory calibration against the thermogravimetric method, which is the standard for SWC measurement [2,14,24,26,27]. These studies have demonstrated that with soil-specific calibration, the sensor can achieve high accuracy, with reported R^2^ values between 0.85 and 0.98 across various soil textures [6,14,24,28]. The necessity for soil-specific specific calibration is a recurring theme, as the sensor’s response is influenced by soil texture, mineralogy, bulk density, and salinity [3,21,22,24,28,29,30]. While custom linear or polynomial calibration functions yield improved results over manufacturer-provided ones [6,14,22,24,30], some studies note that the sensor may not offer sufficient resolution for applications like precision agriculture, with deviations of up to 12% from actual moisture content being reported [6,10]. Research has also highlighted significant sensor-to-sensor variability, especially at higher moisture levels, reinforcing the need for individual sensor calibration [14,24,31].

However, a significant research gap remains in the characterisation of these sensors. The current literature predominantly relies on soil-based calibrations, which can be confounded by variables such as soil texture, soil composition, air gaps, and density variations. A more robust approach, as advocated by Jones et al. [32], involves evaluating electromagnetic sensors in homogeneous fluids of known permittivity [7,32,33]. This standardised methodology allows for the systematic assessment of sensor response across a relevant range of dielectric permittivities in a controlled setting, and it has been widely adopted in many similar studies [8,9,31,33,34,35,36]. The application of this fluid-based characterisation to the SEN0193 sensor is absent in existing studies [3,30], to the best of our knowledge. A second challenge is the lack of fully reproducible, open-source datalogging systems. Beyond calibration issues, a critical barrier to the wider adoption of low-cost soil moisture sensors lies in the lack of reproducible and transparent datalogging solutions. While many studies [16,17,24,29,37,38,39] present custom-built loggers, the absence of detailed documentation and open designs prevents exact replication. This lack of reproducibility limits the comparability of monitoring results across studies and hinders the development of consistent, large-scale sensor networks. For hydrological and ecological applications, where scientific credibility depends on transparent methods and traceable data, open-source hardware and software provide an essential foundation [7,9,10,11,12]. By ensuring that both the hardware design and control software are openly available and fully documented, low-cost monitoring can advance from ad hoc, isolated prototypes to robust and transparent infrastructures, enabling reliable use in research and practice.

To address these gaps, this study has two primary aims. First, we subject the LCSMS to the standardised testing methodology outlined by Jones et al. [32], using reference fluids to relate the sensor’s output to the permittivity ($\epsilon_{s}$). This approach allows for the formulation of a sensor-unit-specific calibration function, applicable to the LCSMS, providing a robust and comparable assessment of the sensor’s performance. The assessment is put into context by including the comparison with three widely used commercially available sensors. Second, we introduce a fully open-source, modular, off-grid-ready datalogger design, along with the adaptation of the LCSMS and a soil-temperature sensor, utilising readily available and well-documented consumer microelectronics. Beyond these immediate technical objectives, this work seeks to leverage recent advancements in *IoT devices* and *smart things* for hydrological monitoring applications. By developing and disseminating open-source hardware and software designs, we aim to lower the barrier of entry for hydrologists interested in the design and assembly of monitoring systems, potentially fostering community-driven improvements and adaptations to accommodate a wider range of sensors, use-cases, and communication protocols such as LoRa(-WAN). The overarching goal is to present a completely open-source and modular soil moisture monitoring station—picoSMMS—whose design can be adapted for various hydrological monitoring applications, and to provide a comprehensive preliminary assessment through field deployment and validation.

## 2. Materials and Methods

This section is organised into four subsections, which describe the data acquisition hardware, the control software, the experimental setup for sensor calibration and validation, and the field installation. While this section provides a concise overview, comprehensive technical details are provided in the appendices.

### 2.1. Hardware

This sub-section details the development of the picoSMMS, a low-cost, off-grid monitoring station designed to measure and record soil moisture and temperature. The core of the picoSMMS is a data acquisition and logging system, hereafter referred to as the picoLogger. The picoLogger is designed around a central compute module with several peripheral components to facilitate the measurement and recording of soil moisture and temperature data. The design prioritises the use of readily available, well-documented, and affordable components to ensure high reproducibility. A list of materials, along with the specifications of each component and a rough estimate of their cost, is provided in Section A.1.

The central compute module of the picoLogger is the Raspberry Pi Pico, hereafter referred to simply as Pico. It was selected for its low cost, extensive documentation, wide consumer adoption, and robust support for various communication protocols, including I2C, SPI, and UART, which are essential for interfacing with the system’s peripheral modules.

The LCSMS, as seen in Figure 1, constitutes the main sensor. It provides an analog voltage output, which requires conversion to a digital signal for processing. While the Pico features a built-in Analog-to-Digital Converter (ADC), its resolution is limited, and its accuracy is susceptible to noise from the Pico’s switching-mode power supply. To ensure high-precision measurements [40], an external 16-bit ADC, the ADS1115, is employed. This module communicates with the Pico via the I2C protocol and provides four input channels. Three channels are dedicated to reading data from the LCSMS, while the fourth is used for monitoring the system’s battery voltage via a voltage divider circuit.

Accurate timestamping of the collected data is critical for temporal analysis. To this end, a DS3231 Real-Time Clock (RTC) module is integrated into the picoLogger. The DS3231, which also communicates via I2C, provides precise and temperature-compensated timekeeping and has a backup battery, ensuring data integrity even in the event of a main power failure.

For data storage, the 2 MB of onboard flash memory on the Pico is insufficient for long-term, high-frequency data logging. Therefore, a microSD card module is incorporated to provide expandable storage capacity, allowing for extended, autonomous data collection campaigns. This module interfaces with the Pico via the SPI protocol.

The system is powered by a 3000 mAh LiPo battery, which is charged by a 1 W solar panel via a dedicated solar charger module (DFRobot DFR0264). To ensure stable and safe operation, the power management circuitry includes a Schottky diode for reverse current protection and a Zener-clamp circuit to regulate the solar panel’s output voltage. An IRFZ44N MOSFET (Infineon Technologies, Prague, Czechia) is used to switch power to the peripherals, minimising idle power consumption. This power system is designed for perpetual, off-grid operation; a detailed power budget analysis is provided in Appendix A.

In addition to the core components, several peripherals are integrated for user interaction and diagnostics. These include an RGB LED for visual status indication, two toggle switches for power and function control, and a UART interface for accessing operational logs.

The picoLogger, its current version as seen in Figure 2, is designed to interface with two types of sensors: the LCSMS for measuring soil moisture and the DS18B20 (hereafter referred to as STS), as seen in Figure 3, for measuring soil temperature. The exposed electronics on the LCSMS units are waterproofed using a custom 3D-printed housing filled with epoxy resin. The STS units are pre-packaged in a waterproof metal casing. All external components are connected to the main unit via robust, waterproof AMP SuperSeal connectors.

The entire picoLogger is housed within a custom-designed, 3D-printed enclosure. The internal components are assembled on two perfboards for a compact and organised layout. The cost of materials for the entire station was less than UDS 100. Further details on the physical assembly and wiring are available in Appendix A.

### 2.2. Software

The picoLogger is programmed in MicroPython, an efficient implementation of Python 3 for microcontrollers. The software is designed for modularity, robustness, and low power consumption, ensuring reliable long-term data collection in off-grid environments. The operational logic is governed by a main control script that cycles through a series of states: activation, data recording, state update, and deactivation, followed by a deep-sleep period to conserve power.

The software architecture is divided into three main parts: a high-level configuration file, the main application logic, and a library of hardware drivers. The configuration file allows for easy modification of system parameters, such as sensor pin assignments and data acquisition intervals, without altering the core logic. The main application handles the initialisation of peripherals, the data acquisition and storage sequence, power management, and scheduling of sleep cycles. The driver library contains modules for interfacing with the specific hardware components, such as the ADS1115 ADC, DS3231 RTC, and the microSD card, sourced from open-source contributors. A detailed description of the software components, their interactions, and the data storage format is provided in Appendix B. The complete source code is open-source and available in the https://doi.org/10.5281/zenodo.17436153.

### 2.3. Experimental Setup

To assess the accuracy and reliability of the LCSMS, a rigorous calibration and validation procedure was conducted in a controlled laboratory environment following the guidelines proposed by Jones et al. [32] and closely modelled after Bogena et al. [8].

Liquids with well-defined $\epsilon_{s}$ values ranging from 1.0 (air) to approximately 80.0 (deionised water), hereafter referred to as “reference media”, were prepared using approximately 15 L of 2-isopropoxyethanol (i-C3E1; Art.-Nr. 8706.2, Carl Roth GmbH, Karlsruhe, Germany), 10.5 L of deionised water, and soda lime glass beads (type: Silibeads 4501, Sigmund Lindner GmBH, Warmensteinach, Germany) with a grain size of 0.25–0.5 mm. The glass beads, when thoroughly shaken to achieve good packing, exhibit a permittivity of 3.34 at 25 °C and a porosity of 0.38 [8]. These media, detailed in Table 1, were designed to cover the full spectrum of permittivities with particular focus on the range encountered in typical inorganic-heavy soils. The permittivities of each medium were measured using Time Domain Reflectometry (TDR; CS645 from Campbell Scientific Inc., Logan, UT, USA) probes at the respective experimental temperatures and are hereafter referred to as “reference permittivity or ($\epsilon_{r}$)”. Additional equipment included magnetic stirrers with a hot-bed, a sieve shaker, thermometers, and standard laboratory glassware.

The calibration and validation experiments were conducted in two phases. The first experiment, carried out in 2022 as part of a previously unpublished study, subjected a diverse set of commercial sensors—SMT100 (TRUEBNER GmbH, Neustadt, Germany), ML2 ThetaProbe (Delta-T Devices Ltd., Cambridge, UK), and PR2/6 ProfileProbe (Delta-T Devices Ltd.), hereafter referred to as “benchmark sensors”—to validate across reference media at 17 ± 1 °C. A subsequent experiment in 2025 focused on calibration and validation of the LCSMS at 25 ± 1 °C, employing a similar but refined protocol that included additional intermediate reference media within the critical permittivity range (2.5 to 35.5) typical of inorganic-heavy soils. The 2022 benchmark sensor dataset provided comparative context for evaluating the calibrated LCSMS performance.

The experimental apparatus comprised the following components. Two LCSMS units paired with a soil temperature sensor (STS; DS18B20) were interfaced with the picoLogger for data acquisition. Six SMT100 sensors, five ML2 ThetaProbe units (), and three PR2/6 ProfileProbe units were read, using a fixed immersion depth, via an Arduino Uno-based data acquisition system. Three TDR probes provided reference permittivity measurements and were read manually through a CR6 datalogger. The measurements, excluding the LCSMS temperature-sensitivity tests, were conducted in ten containers (6.4 dm^3^ HDPE bottles and PVC containers) selected to be sufficiently large to minimise edge effects. For the LCSMS temperature-sensitivity tests, sensors were positioned diagonally in a beaker with the sensing side facing downwards, as the sensor’s limited sensing volume minimises fringe effects; notably, approximately 5 mm of the bottom-most portion of the LCSMS, where no capacitive traces are present, does not contribute to sensing. Examples of the measurement setup are shown in Figure 4 and Figure 5.

A standardised protocol was followed for all measurements to ensure consistency and repeatability. Prior to each measurement, liquid solutions were thoroughly stirred, and the glass bead medium was homogenised using a sieve shaker for 120 s.

To investigate the relationship between the sensor’s output and permittivity, each probe was fully immersed in the centre of its container as their readings were recorded. PR2 ProfileProbe measurements were taken for each of the six rings individually by adjusting the immersion depth. The temperature of the medium was recorded concurrently for all measurements. To prevent cross-contamination, sensors were rinsed with deionised water after each measurement, and containers were kept closed to minimise evaporation. For the LCSMS, two sets of 25 readings were recorded with 0.1 s intervals between readings and 15 s intervals between sets. The measurements of LCSMS were taken at a stable temperature of 25 ± 1 °C, while that of benchmark sensors were taken at a stable temperature of 17 ± 1 °C. For the benchmark sensors, 20 readings were taken in each medium at 15 s intervals over a 300 s period.

To investigate temperature effects of LCSMS, LCSMS and STS were immersed in selected reference media (M3, M4, M5, and M6) that was initially cooled and then subject to gradual heating on a hot-plate along with continuous stirring using a magnetic stirrer. Readings from LCSMS were recorded for every 0.1 °C change in temperature; initial and final readings were discarded to account for thermal equilibration time.

The raw value from an LCSMS reading is a 16-bit signed integer reading from the ADC, henceforth referred to as “raw sensor value” of RSV that corresponds to the voltage output of the LCSMS sensor, henceforth referred to as “sensor/LCSMS output voltage” or VLT. Given that the ADS (ADS1115) was configured with a gain of 4.096 V, as in this case, the relationship between the LCSMS’ output voltage and the raw sensor value is given by Equation (Equation 1):(1)$$\mathrm{VLT}=\frac{\mathrm{RSV}}{32767}\cdot 4.096$$
where 32,767 is the maximum 16-bit signed integer value.

The procedure for the LCSMS was conducted using two units: one for model calibration and the other for independent validation. The objective was to establish a model, $\epsilon_{p}=M\left(NSV,TMP\right)$, that predicts the permittivity of the media the sensor is exposed to, from the normalised sensor value (NSV; a normalised value of the sensor output as described in Equation (Equation 2)) and the temperature of the said media (TMP, in °C). The permittivity predicted from such models shall henceforth be referred to as “predicted permittivity” or $\epsilon_{p}$. The raw sensor value was normalised for each sensor using measurements in air and deionised water, effectively scaling each sensor’s output to a common reference frame from 0 (air) to 1 (water), with normalised values increasing with increasing permittivity, as shown in Equation (Equation 2).(2)$$\mathrm{NSV}=\frac{\mathrm{RSV}-{\mathrm{RSV}}_{\mathrm{air}}}{{\mathrm{RSV}}_{\mathrm{water}}-{\mathrm{RSV}}_{\mathrm{air}}}$$
where ${\mathrm{RSV}}_{\mathrm{air}}$ is the raw sensor value in air, and ${\mathrm{RSV}}_{\mathrm{water}}$ is the raw sensor value when fully immersed in deionised water.

The operating principle of LCSMS is based on a capacitor formed by two PCB traces acting as plates with the surrounding medium serving as the dielectric. The sensor’s output is therefore sensitive to the physical dimensions of these traces, including their thickness, spacing, and coating. Due to the low-cost manufacturing process, dimensional inconsistencies are anticipated, leading to non-negligible variability between individual sensor units. While this variability was not formally quantified in this study, the normalisation procedure establishes a common measurement scale. To assess the effect of normalisation, similar models were also trained using the sensor’s output voltage and corresponding temperature, henceforth referred to as “VLT-trained models” whose performance was compared with that of models trained on normalised sensor values, henceforth referred to as “NSV-trained models”.

Several modelling approaches were evaluated to predict permittivity from the normalised sensor value and temperature, including Linear, Polynomial, Random Forest, Gradient Boosting, and Support Vector Regression. The training dataset comprised 12,075 samples with feature inputs being (1) voltage reading (VLT) and temperature (TMP), and (2) normalised sensor value (NSV) and temperature (TMP). To focus the calibration on the permittivity range most relevant to typical inorganic-heavy soils, a weighting scheme was employed during model training, prioritising accuracy within the 2.5 to 35.5 permittivity range by assigning a weight of 1.0 to samples within this range and 0.1 to samples outside it. This resulted in 11,875 of the 12,075 training samples being given higher priority. For validation, the second sensor was subjected to all reference media and its output was recorded, generating 83 unique test samples. As a validation strategy, resulting measurements were used to predict permittivities with the trained models, and the predicted permittivities were then compared to the reference permittivities, with results presented in the subsequent sections.

Similarly, for the benchmark sensors, a two-point normalisation procedure was applied. Given that the deviation in the measured variable can be assumed linear, measurements in air and deionised water were used to correct the sensor outputs. This approach preserved the manufacturer’s conversion from the primary measured variable to permittivity. For the ThetaProbe units and PR2/6 ProfileProbe units, voltage values were corrected, while for the SMT100 sensors, the permittivity reported by the sensor was treated as the primary variable and adjusted accordingly to preserve the manufacturer’s device-specific conversion. Ref. Francke and Terschlüsen [41] provides a detailed description of this procedure, along with an R-package to perform such recalibrations. The permittivities reported by benchmark sensors using the manufacturer-provided calibration shall henceforth be referred to as “measured permittivity”.

For data management, raw data gathered by the Arduino logger from the benchmark sensors, stored on its SD card, was exported to Excel, organised by date, time, and sensor ring, and merged with manually recorded temperature and medium information. For the LCSMS experiments, individual data files were merged into a single pandas dataframe containing sensor ID (SID), temperature (TMP), reference media (SOL), reference permittivity (EPS), and raw sensor value (RSV), which was then exported to a single CSV file.

The complete data processing and analysis workflow (as a IPython Notebook), along with the raw datasets themselves, are available in the https://doi.org/10.5281/zenodo.17436153.

### 2.4. Field Installation

To demonstrate the operational feasibility of the picoSMMS and demonstrate a preliminary proof-of-concept, initial field deployments of the first-generation device were conducted in the Rokytka Catchment (Šumava mountains, Czech Republic; Figure 6) and the Berembadi catchment (Chamarajanagar, Karnataka, India; Figure 7). These deployments serve as a preliminary assessment of real-world performance and durability, rather than a formal field validation of the LCSMS sensor. The rationale for such a test was that despite the limitations in direct comparison between co-located sensors arising from soil-heterogeneities and in varied approaches of converting the recorded sensor values (relating to $\epsilon_{s}$) to VWC, such comparisons are helpful in assessing the VWC trends over longer time periods.

However, these deployments revealed critical vulnerabilities in the design of picoSMMS. In Rokytka, the device failed after approximately three weeks due to voltage spikes that reset the RTC. A similar issue affected the Berembadi station, which produced faulty readings after one week and failed completely after four weeks. This station also experienced a sensor failure at a 50 cm depth, attributed to an unreliable connector. These early field experiences underscored the necessity for more robust power management and connectors and raised questions on the sensor’s temperature-dependent response. A detailed account of the field installations and the iterative hardware improvements is provided in Appendix C.

The insights gained from these early tests informed the development of a second-generation picoSMMS (picoSMMS v2), presented in the sections above, which featured significant hardware revisions to improve robustness. These improvements included enhanced power protection circuitry (a Zener-clamp to limit solar panel voltage), upgraded waterproof AMP SuperSeal connectors, a more reliable and temperature-compensated DS3231 RTC, and the integration of dedicated soil temperature sensors. picoSMMS v2 has only been subject to laboratory tests as of yet, and field deployments will be carried out in the near future.

## 3. Results

### 3.1. Experimental Results

#### 3.1.1. Assessing the Individual Impacts of Permittivity, Temperature, and Normalisation

The sensor output voltage (VLT) from the LCSMS is presented alongside the normalised sensor values (NSV) across the tested permittivities to demonstrate the sensor’s response characteristics. Figure 8 presents both the training and testing datasets, visualising the raw voltage output and the normalised raw sensor values against reference permittivities measured at 25 ± 1 °C.

The figure reveals a largely linear relationship between voltage and permittivity within the target range (2.5 to 35.5), with the sensor demonstrating clear sensitivity across different permittivity levels. Significant deviation from linearity is observed beyond a permittivity of 35.5, which exceeds the physically meaningful range for typical mineral soils at full saturation. The normalisation aims to reduce scatter in the data and produces a more consistent relationship between sensor output and permittivity across different sensors, with values rising with rising permittivities. While the normalisation shows limited if not negligible effects in the limited sensor ensemble tested here (two LCSMS units), it addresses an important practical consideration. Manufacturing variability in capacitive sensors, as discussed in Section 2.3, can result in non-negligible baseline differences (up to ±0.1 V in our experience with LCSMS units). The manufacturer’s rudimentary calibration approach—immersing the sensor in water (saturation = 1) and exposing it to air (saturation = 0), then linearly interpolating between these bounds—implicitly acknowledges this inter-sensor variability. While our two-sensor dataset provides insufficient statistical power to comprehensively characterise the magnitude and consistency of such variability, the air–water normalisation framework presented here offers a practical pathway toward sensor-unit-specific calibrations. A more robust assessment of inter-sensor variability across a larger sample of LCSMS units would be essential to fully validate the necessity and effectiveness of this normalisation procedure.

To better understand the temperature dependence of the raw sensor response, measurements of the sensor’s voltage output over varying temperature from four representative reference media (M3, M4, M5, and M6) spanning the target permittivity range are examined in detail. Figure 9 illustrates the relationship between the sensor’s output voltage (VLT) and temperature (TMP) for these solutions. Linear regression analysis for each solution reveals a consistent positive temperature coefficient, albeit with varying slopes and intercepts, with the sensor output voltage increasing systematically with temperature for a given permittivity. The temperature coefficients range from 1.54 to 3.14 mV/°C across the examined reference media (M3: 2.45 mV/°C, M4: 3.14 mV/°C, M5: 2.60 mV/°C, M6: 1.54 mV/°C), demonstrating the sensor’s temperature sensitivity across the target permittivity range. This linear temperature dependence, clearly demonstrated by the fitted regression lines, forms the physical basis for including temperature as a predictive feature in the calibration models.

#### 3.1.2. Discussion on the Models

After evaluating several regression models, the Linear Regression and second-degree Polynomial (Polynomial D2) models were selected for their robust performance and adherence to the physical nature of the underlying processes, with the Polynomial D2 taking precedence. The detailed performance metrics of the models are presented in Table 2.

#### 3.1.3. Pre-Normalisation Results

When trained with raw voltage (VLT) and temperature, the resulting sensor-specific calibration equations are:Linear Regression (VLT):(3)$$\epsilon_{s}=142.9424-60.3976\cdot \mathrm{VLT}+0.1738\cdot \mathrm{TMP}$$Polynomial D2 (VLT):(4)$$\begin{array}{cc} \epsilon_{s}=161.7648 & -\;74.5491\cdot \mathrm{VLT}-0.3535\cdot \mathrm{TMP}\\ \; & +\;2.2510\cdot {\mathrm{VLT}}^{2}+0.3001\cdot \mathrm{VLT}\cdot \mathrm{TMP}-0.0028\cdot {\mathrm{TMP}}^{2} \end{array}$$

Figure 10 illustrates how these VLT-trained models predict permittivity across the sensor output range, demonstrating strong performance in capturing the underlying sensor response. It is important to note that these VLT-based calibrations are sensor-specific and would potentially require recalibration for different LCSMS units due to manufacturing variability. Figure 11 compares the VLT-based LCSMS predictions, from the Polynomial D2, with the manufacturer-calibrated permittivities measured from three commercial sensors (SMT100, ML2 ThetaProbe, and PR2/6 ProfileProbe) on the validation dataset, demonstrating that the LCSMS performs competitively with established commercial sensors.

The VLT-based Linear Regression model achieved a mean absolute error of 1.37 ± 1.12 over the target permittivity range with a maximum error of 3.80, while the Polynomial D2 model showed 1.37 ± 1.35 (max: 4.48). When translated to volumetric water content (VWC) using the derivative of the Topp equation [42], the Linear model yielded VWC errors of 0.02 ± 0.01 (max: 0.05), while the Polynomial D2 model achieved 0.02 ± 0.01 (max: 0.03). Detailed error distributions across reference media are presented in Figure 12 and Figure 13.

#### 3.1.4. Post-Normalisation Results

To achieve sensor-unit-specific calibration that could potentially eliminate the need for individual sensor-specific calibrations, models were trained with normalised sensor values (NSV) and temperature. The resulting sensor-unit-specific calibration equations are:Linear Regression (NSV):(5)$$\epsilon_{s}=-4.5380+64.6803\cdot \mathrm{NSV}+0.1738\cdot \mathrm{TMP}$$Polynomial D2 (NSV):(6)$$\begin{array}{cc} \epsilon_{s}=-6.8496 & +\;68.0628\cdot \mathrm{NSV}+0.3792\cdot \mathrm{TMP}\\ \; & +\;2.5815\cdot {\mathrm{NSV}}^{2}-0.3213\cdot \mathrm{NSV}\cdot \mathrm{TMP}-0.0028\cdot {\mathrm{TMP}}^{2} \end{array}$$

These NSV-based calibrations can be applied to any LCSMS unit after simple air and water measurements, potentially eliminating the need for extensive recalibration with reference media for each individual sensor. Figure 14 visualises the NSV-trained models’ predictions against the normalised sensor values. Notably, the Polynomial D2 model more effectively captures the non-linear sensor response outside the target range on the lower side compared to the Linear Regression model. Figure 15 shows a strong correlation between the predicted and reference permittivities for the NSV-based models, with data points tightly clustered around the 1:1 line, particularly within the target permittivity range (2.5 to 35.5). This figure also includes comparison with the benchmark sensors using their normalised (re-calibrated) permittivities, showing the advantage of such a calibration. The NSV-trained Linear Regression achieved a mean absolute error of 1.29 ± 1.07 (max: 3.72) over the target range, and the Polynomial D2 model achieved a mean absolute error of 1.24 ± 1.36 (max: 4.40). The corresponding VWC errors were 0.02 ± 0.01 (max: 0.04) for the Linear model and 0.01 ± 0.01 (max: 0.03) for the Polynomial D2 model. These errors remain acceptable for many hydrological and hydro-pedagogical applications, particularly considering the sensor’s low cost.

It must be noted that the manufacturer-specified VWC ranges for the benchmark sensors are as follows: PR2 ProfileProbe: 0–0.4 [m^3^/m^3^], ML2 ThetaProbe: 0.05–0.6, SMT100: 0–0.6.

The comparison with re-calibrated benchmark sensors demonstrates that when air–water normalisation, as detailed in Francke and Terschlüsen [41], is applied to the benchmark sensors, the performance of ML2 ThetaProbe and the PR2/6 ProfileProbe increase significantly, especially within the target permittivity range. Detailed error distributions across reference media are presented in Figure 16 and Figure 17. The performance gains from the normalisation procedure for the LCSMS with the given sample size (two units) is statistically insignificant. Inter-sensor variability and the effect of normalisation on the performance of the LCSMS remains to be assessed. However, owing to the reasons discussed before, such a pathway offers a physically well-founded pathway for such an assessment. It is important to note that due to the limited sample size of LCSMS units tested (2 units) compared to the benchmark sensors (5 units each), these results should be interpreted with caution. With a larger sample size, the LCSMS may, and likely will, exhibit greater inter-sensor variability whose effects might not be fully addressable even with the normalisation procedure, potentially resulting in larger prediction errors than reported here.

#### 3.1.5. Summary

In summary, the experimental results demonstrate that the LCSMS can be effectively calibrated using both sensor-specific (VLT-based) and sensor-independent (NSV-based) approaches, with the latter being a better option, to establish a common scale if not to potentially tackle inter-sensor variability. Table 2 summarises the validation performance metrics for both the models trained with both feature sets, demonstrating strong predictive performance across the target permittivity range. The choice between VLT-based and NSV-based calibrations depends on whether sensor-specific accuracy or sensor-unit-specific applicability is prioritised. For applications requiring deployment of multiple sensors or where recalibration is impractical, the NSV-based approach offers a compelling balance between accuracy and operational convenience.

Table 3 illustrates the effect of air–water normalisation for the benchmark sensors.

Detailed per-solution predictions and absolute errors for the VLT-based and NSV-based calibrations can be found in Section D.1.

### 3.2. Results from the Field Installation

For the preliminary field deployment, the NSV-based Linear Regression model, developed using the experimental tests as described in the previous section, was selected due to its simplicity and robust performance. The raw sensor values (RSV) from the installed LCSMS units were converted to normalised sensor values (NSV) using measured raw sensor values in air and water for each sensor pre-deployment. The NSV and concurrent temperature data, which were sourced from the internal temperature sensor of the co-located HydraProbe (given the fact that the first version of the picoSMMS installed at this site did not include the STS), were then used to predict the permittivity. Finally, the predicted $\epsilon_{p}$ was converted to VWC using the Topp equation [42], i.e., without any further soil specific adjustments.

Figure 18 presents a time-series comparison of VWC derived from the LCSMS and measurements from a co-located HydraProbe (Stevens Water Monitoring Systems, Inc., Portland, Oregon, USA) sensor. During its first week of operation, the picoSMMS experienced a voltage-spike, which resulted in faulty readings. A second such voltage-spike after about a month of operation resulted in a fatal fault of the station. A more detailed account can be found in Appendix C. The plot reveals a close correspondence between the two sensors, effectively capturing diurnal soil moisture dynamics during a period with no significant precipitation. Interestingly, while the Linear Regression model showed poorer performance in the laboratory tests for the permittivity ranges observed in the field, it yielded VWC predictions closer to the HydraProbe data than the Polynomial D2 model, as seen in Figure A1 in Section D.2. The data demonstrates the LCSMS’s capability to resolve fine-scale temporal variations, such as diurnal cycles. Observed discrepancies between the sensors may be attributed to inherent sensor inaccuracies or spatial heterogeneities in soil properties, potentially arising from natural variability or installation-induced disturbances [7,29,31,38,43]. The limitations of this preliminary field deployment and directions for comprehensive field validation are discussed in detail in Section 4.

## 4. Discussion

This study was motivated by the need for affordable, reliable soil moisture monitoring to capture the high spatio-temporal variability of this critical variable, aiming to transfer technological gains from the burgeoning fields of IoT and consumer electronics to hydrological science. In the spirit of open science, and in the hope of contributing a framework for open-source hydrological monitoring systems for other interested researchers to build upon, the natural course of action was to document the effort well and share the findings, Thus, the primary objectives were (1) to develop a fully open-source, low-cost soil moisture station (picoSMMS) and (2) to conduct a robust, sensor-unit-specific calibration and validation of a popular low-cost capacitive soil moisture sensor (LCSMS; DFRobot SEN0193) by relating its raw output to the bulk static relative dielectric permittivity ($\epsilon_{s}$). Additionally, it was also of interest to demonstrate a proof-of-concept deployment and share findings therefrom. The development of the picoSMMS, built with accessible consumer electronics and programmed in MicroPython, successfully demonstrates a viable pathway for creating affordable, reproducible, and customisable tools for hydrological monitoring. Unlike proprietary “black-box” systems, the open documentation of both hardware and software ensures that raw signals, calibration procedures, and processing workflows are fully traceable. This transparency is essential not only for scientific credibility but also for fostering collaborative improvements by the research community and enabling adoption in citizen science initiatives [38].

The laboratory experiments, conducted under non-conducting and non-relaxing (NC-NR) conditions Jones et al. [32], i.e., disregarding any effects of changes in electrical conductivity (ergo salinity) under varying temperature-conditions, provided a fundamental evaluation of the LCSMS’s performance. By calibrating the sensor against reference media with known permittivities at various temperatures, we established two effective models—Linear Regression and second-degree Polynomial (Polynomial D2)—that relate the sensor’s normalised output and corresponding temperature to $\epsilon_{s}$. Such a characterisation, while incomplete, owing to the fact that it disregards the effects of varying electrical conductivity, i.e., salinity, presents a viable methodology for the utilisation of LCSMS for several use-cases where the salinity of the soil does not significantly vary. Furthermore, it also forms a strong basis for further studies to characterise the LCSMS under varying electrical conductivity.

The benchmarking against commercial sensors provided crucial context for the LCSMS’s performance. It demonstrated that, particularly considering its low cost, its accuracy is acceptable for many hydrological and hydro-pedagogical applications. The results showed that the LCSMS is competitive with the ML2 ThetaProbe, significantly outperforms the PR2/6 ProfileProbe (which showed poor performance with manufacturer settings), but is less accurate than the SMT100. However, it must be emphasised that these comparisons are based on a limited sample size (2 LCSMS units versus 5 units of each benchmark sensor), and with a larger ensemble, the LCSMS may, and likely will, exhibit greater inter-sensor variability whose effects might not be fully addressable even with the normalisation procedure, potentially resulting in larger prediction errors. This positions the LCSMS as a cost-effective alternative for applications where high-density spatial monitoring is more critical than single-point precision. Notably, our tests demonstrate that the performance of the ML2 ThetaProbes, and especially of the PR2/6 ProfileProbe, can be considerably improved at higher permittivity values by applying a similar air–water normalisation (using air and water measurements) as was done for the LCSMS. When this normalisation is applied to the benchmark sensors, their performance increases significantly, particularly within the target permittivity range.

The field deployment, though brief, provided a valuable preliminary proof-of-concept demonstration. The picoSMMS successfully captured diurnal soil moisture dynamics, with VWC predictions from the Linear Regression model closely tracking data from the co-located HydraProbe sensors. However, it is important to emphasise that this deployment does not constitute a formal field validation of the LCSMS sensor. A robust field validation would require careful consideration of probe-specific characteristics such as sensitive volume, installation depth, and soil-sensor contact conditions to ensure equivalence between co-located sensors. These factors can significantly influence measurement comparability and were not rigorously controlled in this preliminary deployment. One must also note the limitations of using a generic, soil-agnostic petrophysical model like the Topp equation to convert permittivity to VWC, despite a pragmatic choice for a quick and rough assessment, as it may not perfectly represent the specific soil characteristics at the field site. The failure of the field station due to voltage spikes also serves as a critical lesson, underscoring the need for more robust power protection circuits in future iterations of the picoSMMS design.

The air–water normalisation procedure deserves particular attention, as it addresses an important practical consideration. Manufacturing variability in capacitive sensors can result in non-negligible baseline differences (up to ±0.1 V in our experience with LCSMS units). The manufacturer’s rudimentary calibration approach—immersing the sensor in water (saturation = 1) and exposing it to air (saturation = 0), then linearly interpolating between these bounds—implicitly acknowledges this inter-sensor variability. While our two-sensor dataset provides insufficient statistical power to comprehensively characterise the magnitude and consistency of such variability, the air–water normalisation framework offers a physically well-founded pathway for establishing a common scale if not to potentially tackle inter-sensor variability, and it underlines the need for a more robust assessment of inter-sensor variability across a larger sample of LCSMS that would be essential to fully validate the necessity and effectiveness of this normalisation procedure.

Despite these promising results, this study has several limitations that highlight directions for future research: (1) the brief field deployment, while successful in demonstrating functionality, was insufficient to meaningfully assess the long-term performance and durability of the picoSMMS across varying seasons and weather conditions; (2) while our laboratory experiments focused on characterising the LCSMS’ performance across varying permittivities and temperatures, its characterisation for varying salinity was left unexplored; (3) the limited sensor ensemble (just two LCSMS units) prevented a comprehensive assessment of inter-sensor variability and the effectiveness of the normalisation procedure across a larger population of sensors.

The open-source nature of the picoSMMS invites community collaboration for expanding its capabilities, such as integrating wireless communication (perhaps with LoRaWAN) or support for other sensors, to further advance the creation of accessible and powerful tools for hydrological monitoring.

## 5. Conclusions

This study developed a fully open-source, low-cost, off-grid soil moisture monitoring station (picoSMMS) and conducted a robust, potentially sensor-unit-specific calibration of the LCSMS (DFRobot SEN0193) by relating its raw output to bulk static relative dielectric permittivity ($\epsilon_{s}$). The picoSMMS, built with readily available consumer electronics and programmed in MicroPython, ensures full transparency of both hardware and software, representing a significant step towards democratising access to reliable soil moisture monitoring tools and facilitating reproducibility and community-driven improvements.

Laboratory calibration under controlled non-conducting, non-relaxing conditions yielded two effective models—Linear Regression and Polynomial D2—with acceptable prediction errors within the target permittivity range typical of mineral soils. Benchmarking revealed that the LCSMS performs competitively with the ML2 ThetaProbe, significantly outperforms the PR2/6 ProfileProbe, but is less accurate than the SMT100, positioning it as a cost-effective alternative for applications where high-density spatial monitoring is prioritised over minute soil moisture dynamics, providing a valuable proof-of-concept. However, these comparisons are based on a limited sample size (2 LCSMS units versus 5 units of each benchmark sensor), and with a larger ensemble, the LCSMS may, and likely will, exhibit greater inter-sensor variability whose effects might not be fully addressable even with the normalisation procedure, potentially resulting in larger prediction errors than reported here.

The air–water normalisation framework offers a practical pathway toward sensor-unit-specific calibrations, addressing manufacturing variability that can result in non-negligible baseline differences. While the modest effects observed in our limited two-sensor ensemble prevent definitive conclusions about inter-sensor variability, the approach is physically well-founded. A robust assessment across a larger sample of LCSMS units would be essential to fully validate the effectiveness of this normalisation procedure. Notably, applying similar normalisation to benchmark sensors demonstrated significant performance improvements, particularly for the ML2 ThetaProbe and PR2/6 ProfileProbe within the target permittivity range, suggesting broader applicability beyond low-cost sensors.

Important usage conditions and limitations must be emphasised: the LCSMS calibration presented is valid for non-saline conditions and performs best in the target permittivity range (2.5–35.5) typical of mineral soils, and may not apply to peat or high-clay soils. The characterisation was conducted at controlled temperatures with temperature-dependency incorporated into the models. Inter-sensor variability remains unaddressed due to the small sensor ensemble, and the brief field deployment was insufficient for long-term performance validation across diverse soil types and environmental conditions. Furthermore, the characterisation under varying electrical conductivity was not addressed in this study.

To overcome these limitations, future work should aim to characterise the LCSMS under varying electrical conductivity, assess inter-sensor variability across larger sensor populations, and conduct long-term field validation in diverse soil types and seasonal conditions. Beyond addressing current limitations, the open-source framework offers opportunities for broader advancements, including integration of wireless communication protocols such as LoRaWAN and adaptation for multi-sensor hydrological monitoring applications.

These findings highlight the potential of transferring advances from consumer electronics and IoT domains to establish transparent, scalable, and cost-effective infrastructures for hydrological monitoring. This work demonstrates a viable pathway for creating affordable, reproducible, and customisable tools while promoting open-science principles, encouraging continued exploration and integration of low-cost sensors and open-source platforms in scientific monitoring with the potential to transform current practices and expand the scope of hydrological observations.

## Figures and Tables

**Figure 1 sensors-25-06907-f001:**
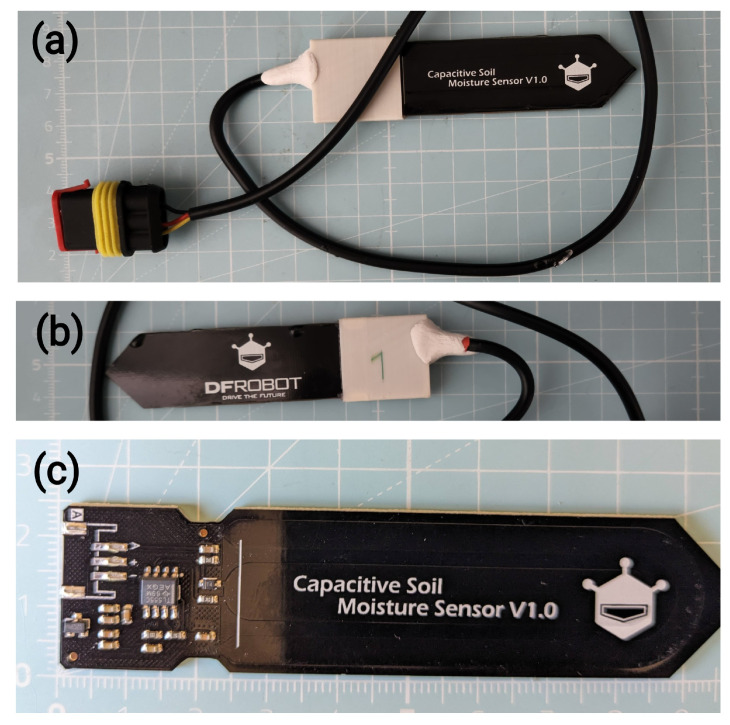
LCSMS: (**a**) front view with the AMP SuperSeal plug visible; (**b**) back view; (**c**) the bare LCSMS sensor, front view, with the capacitor traces and visible. Grid in the background has 1 cm scale.

**Figure 2 sensors-25-06907-f002:**
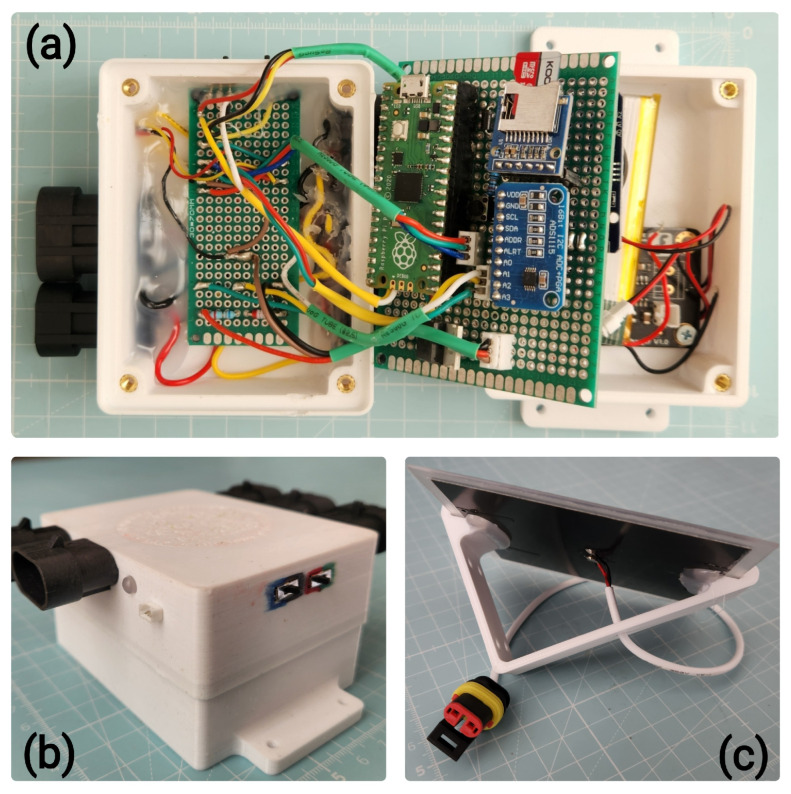
(**a**) Open view of picoLogger v2 showing, from the left, the AMP SuperSeal ports for connecting sensors, and solar panel, power and peripheral management circuitry, the main logic board carrying the MCU, RTC, ADC, and SD card modules, the battery, and the charging module; (**b**) a parametric view of picoLogger v2; (**c**) solar panel, 1 W. Grid in the background has a 1 cm scale.

**Figure 3 sensors-25-06907-f003:**
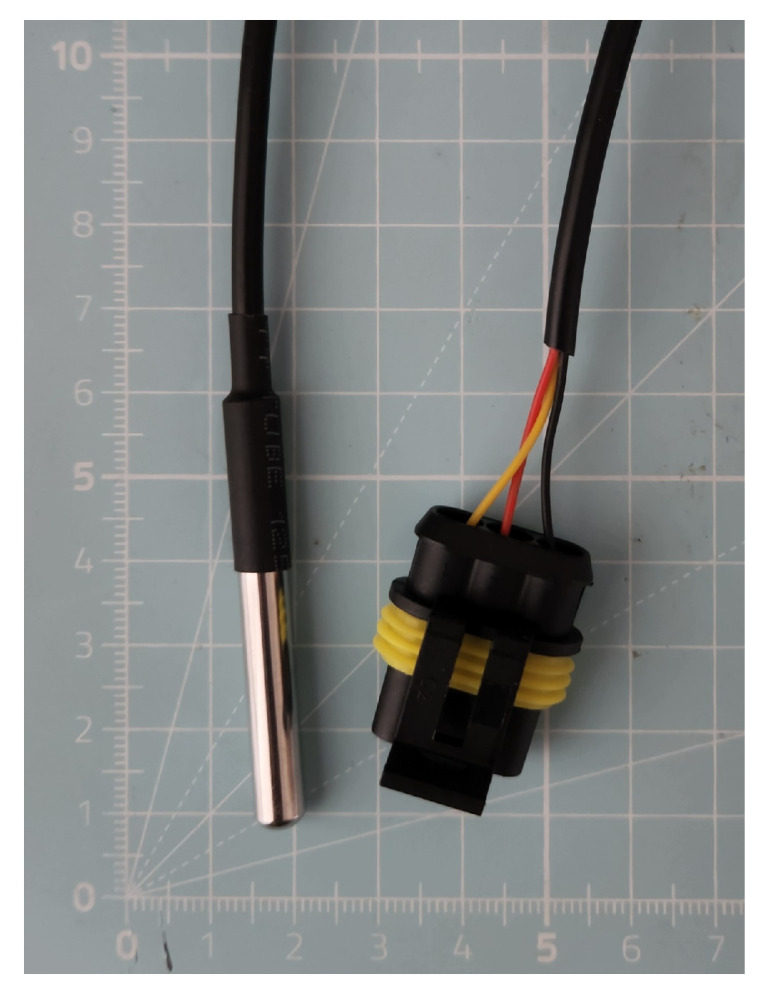
Soil temperature sensor (STS; DS18B20) with a sealed metal casing for waterproofing, with the AMP SuperSeal plug visible. Grid in the background has a 1 cm scale.

**Figure 4 sensors-25-06907-f004:**
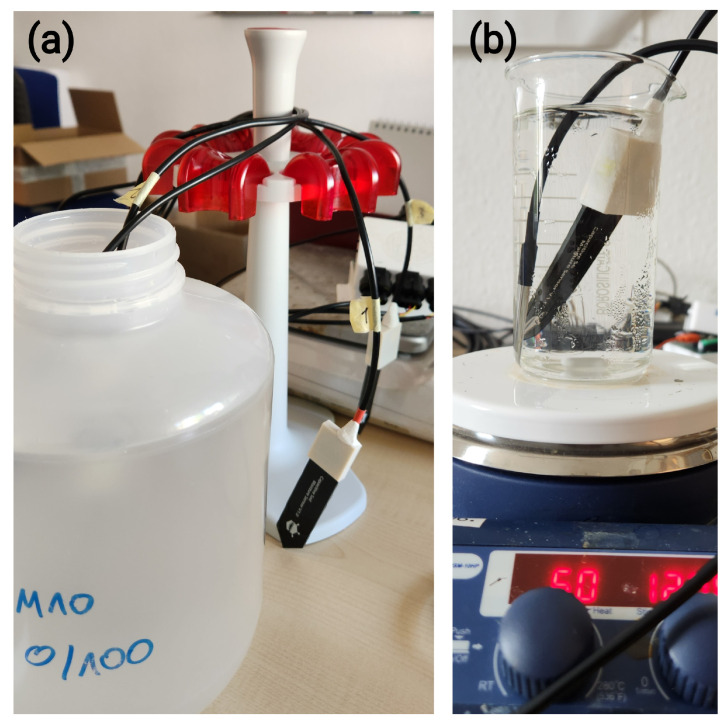
(**a**) Setup used to measure the response of LCSMS, along with the temperature of the solution, across various reference media; (**b**) Setup used to measure the effect of temperature on LCSMS.

**Figure 5 sensors-25-06907-f005:**
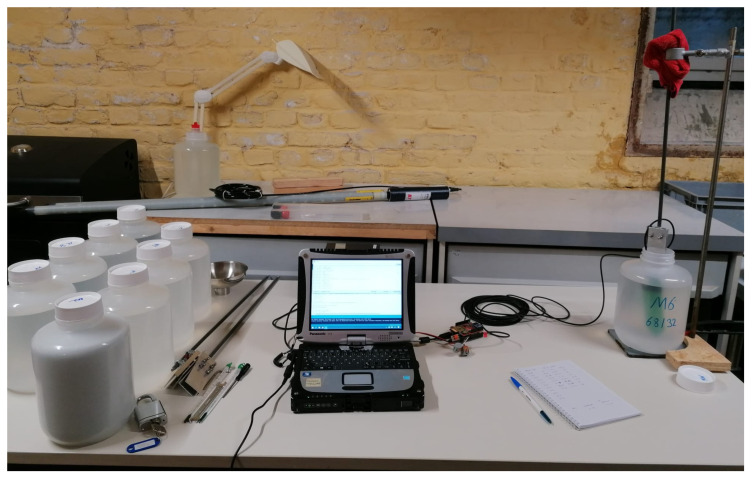
Setup used to measure the response of the benchmark sensors across the reference media.

**Figure 6 sensors-25-06907-f006:**
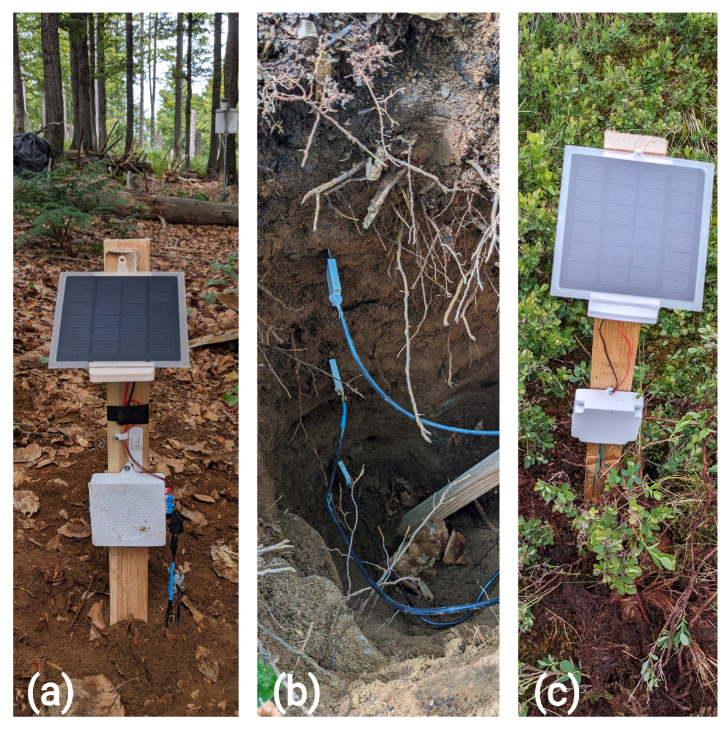
Field installations at Rokytka catchment, Šumava Mountains, Czechia: (**a**) picoSMMS v1 in beech stand; (**b**) LCSMS installed at depths of 15 cm, 40 cm, and 60 cm; (**c**) picoSMMS v0.1 in peat.

**Figure 7 sensors-25-06907-f007:**
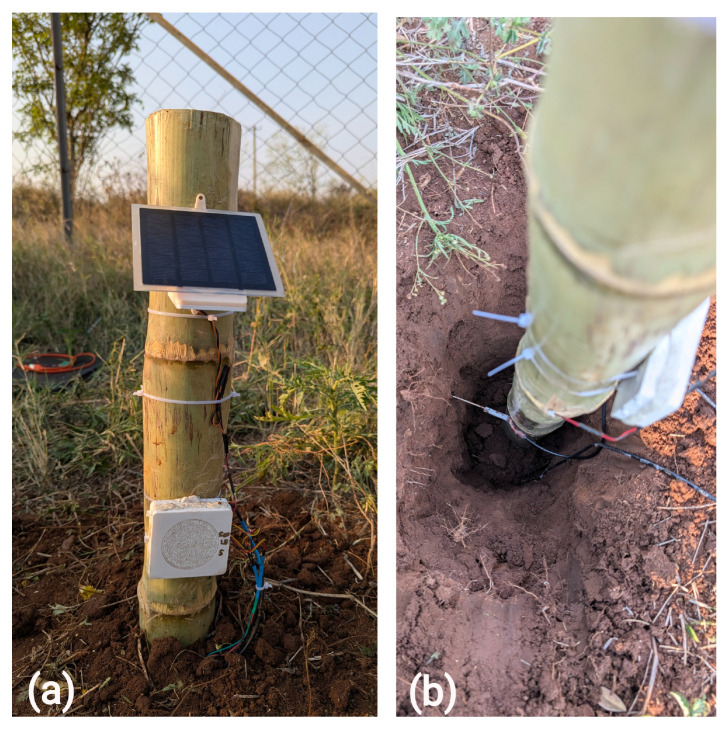
Field installations at Berembadi catchment, Karnataka, India: (**a**) picoSMMS v1 installed; (**b**) LCSMS installed at depths of 5 cm, 15 cm, and 50 cm, similar to the HydraProbes.

**Figure 8 sensors-25-06907-f008:**
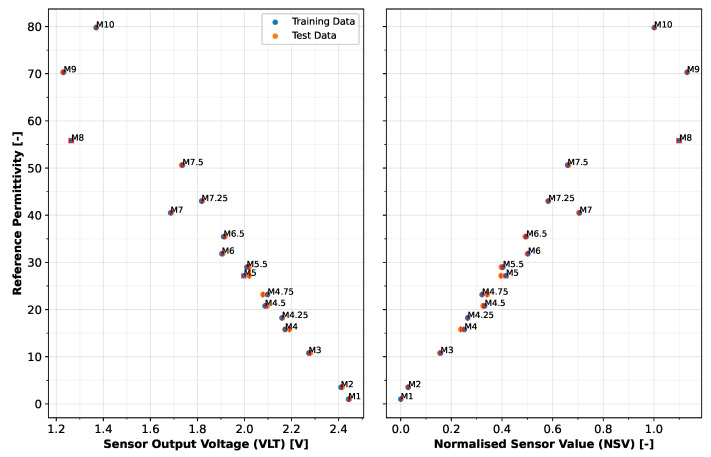
Sensor voltage output (**left**), and normalised sensor value (**right**) against reference permittivities from the training and testing datasets showing the LCSMS response across the permittivity range, measured at 25 ± 1 °C. Error bars, in red, represent the standard deviation of measurements.

**Figure 9 sensors-25-06907-f009:**
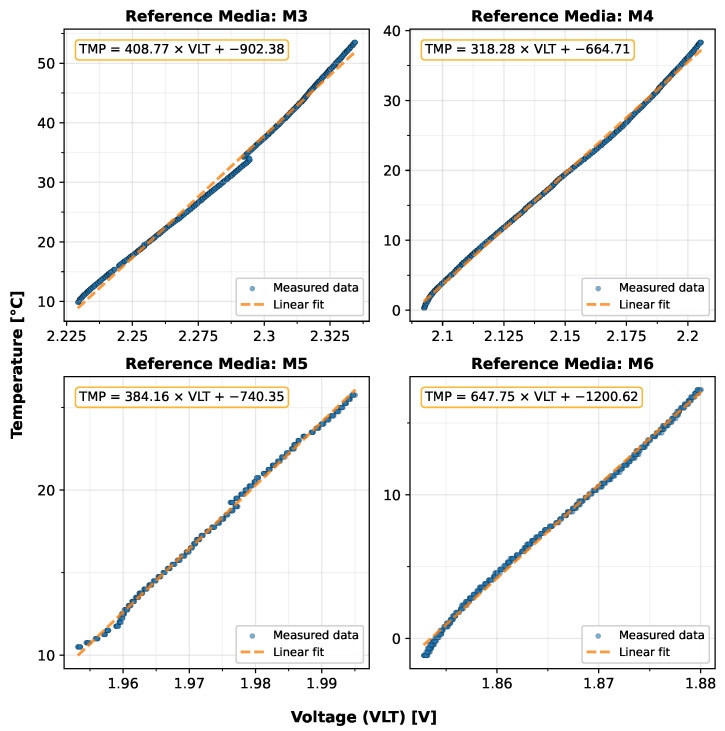
Temperature dependence of the sensor’s voltage output (VLT) for representative reference solutions (M3, M4, M5, M6), from the training dataset, along with the respective equations of the linear regression.

**Figure 10 sensors-25-06907-f010:**
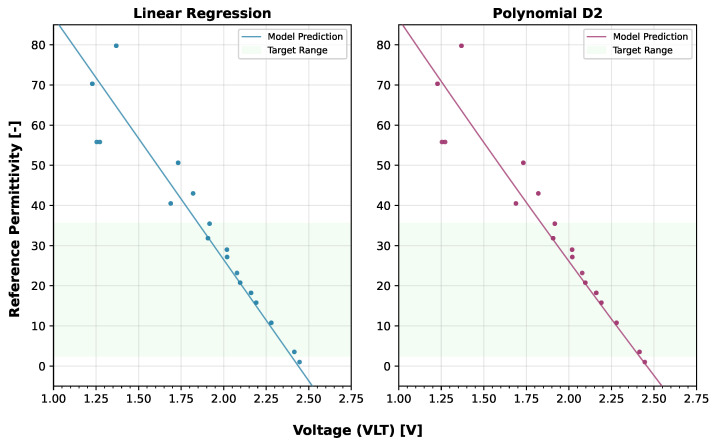
Validation dataset: sensor’s output voltage vs. predicted permittivities at 25 ± 1 °C from Linear and Polynomial D2 models applied to calibrate LCSMS, illustrating model predictions alongside reference permittivities; the scatter shows the sensor’s VLT response against the reference permittivities.

**Figure 11 sensors-25-06907-f011:**
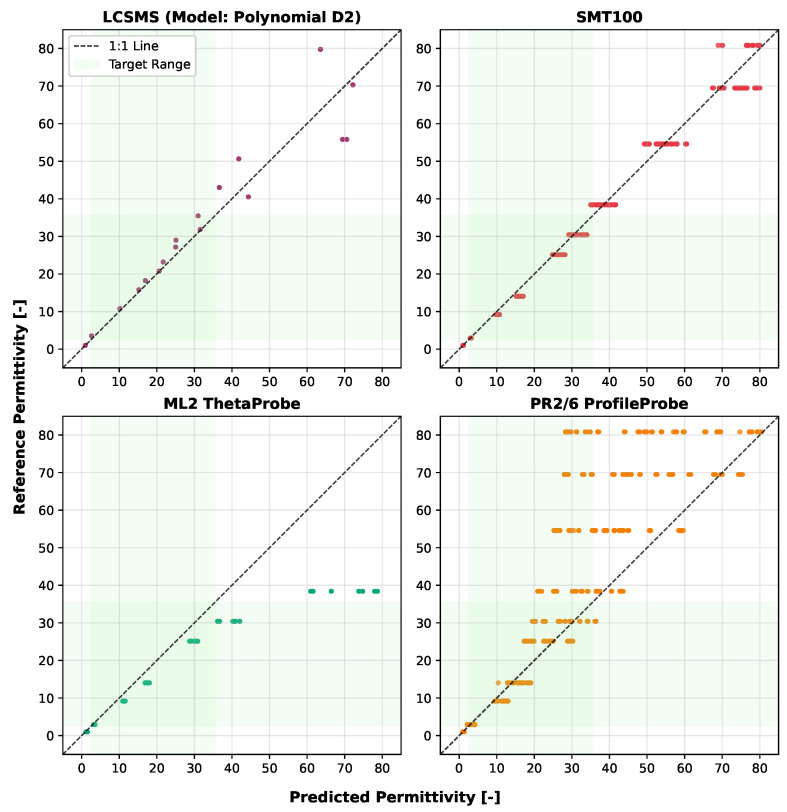
Validation dataset: predicted permittivity (1 LCSMS unit: validation, model: Polynomial D2; VLT-based) vs. reference permittivity at 25 ± 1 °C and measured permittivities from benchmark sensors (5 units each) vs. reference permittivity at 17 ± 1 °C.

**Figure 12 sensors-25-06907-f012:**
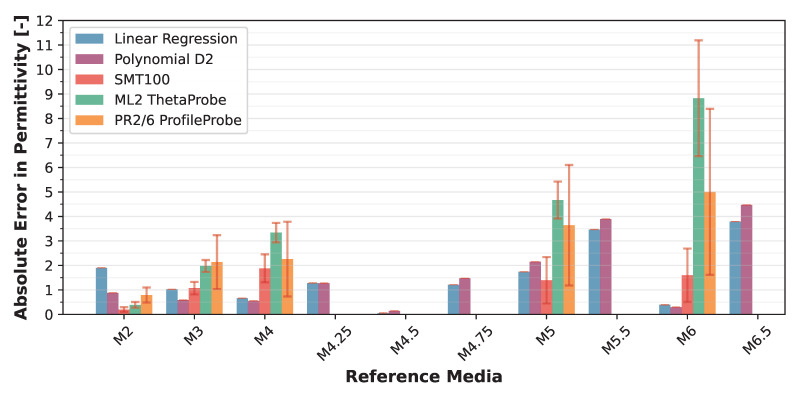
Validation dataset: absolute error (±standard deviation) in predicted permittivity (1 LCSMS unit: validation, model: Polynomial D2; VLT-based) by reference solution (at 25 ± 1 °C) compared with absolute error in predicted permittivity from manufacturer-calibrated benchmark sensors (5 units each).

**Figure 13 sensors-25-06907-f013:**
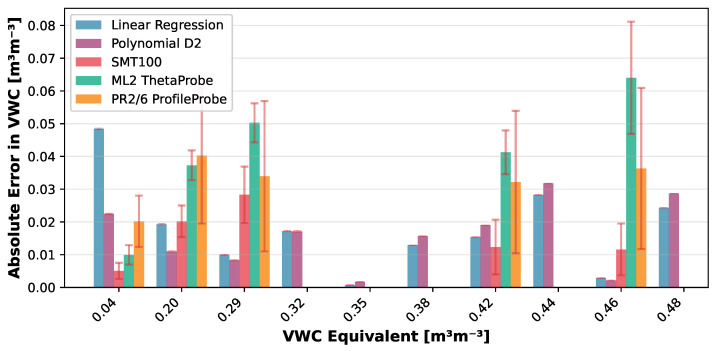
Validation dataset: absolute error (±standard deviation) in VWC inferred via Topp model (1 LCSMS unit: validation, model: Polynomial D2; VLT-based) by reference solution expressed as volumetric water content equivalent compared with absolute error in VWC from manufacturer-calibrated benchmark sensors (5 units each).

**Figure 14 sensors-25-06907-f014:**
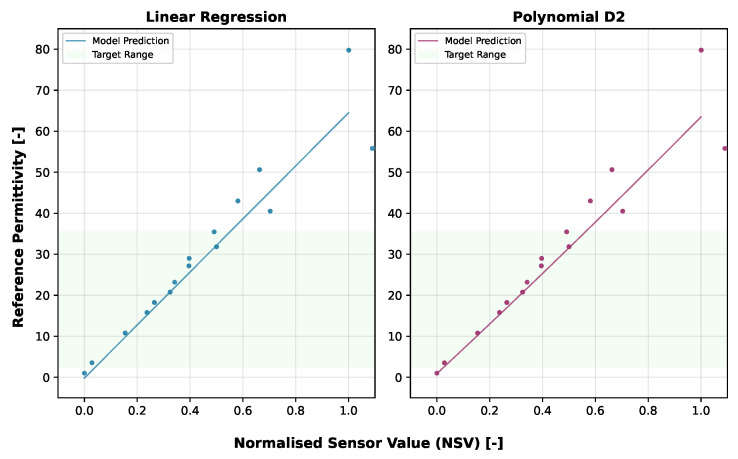
Validation dataset: normalised sensor value vs. predicted permittivities at 25 ± 1 °C from Linear and Polynomial D2 models applied to calibrate LCSMS, illustrating model predictions alongside reference permittivities; the scatter shows the sensor’s NSV response against the reference permittivities.

**Figure 15 sensors-25-06907-f015:**
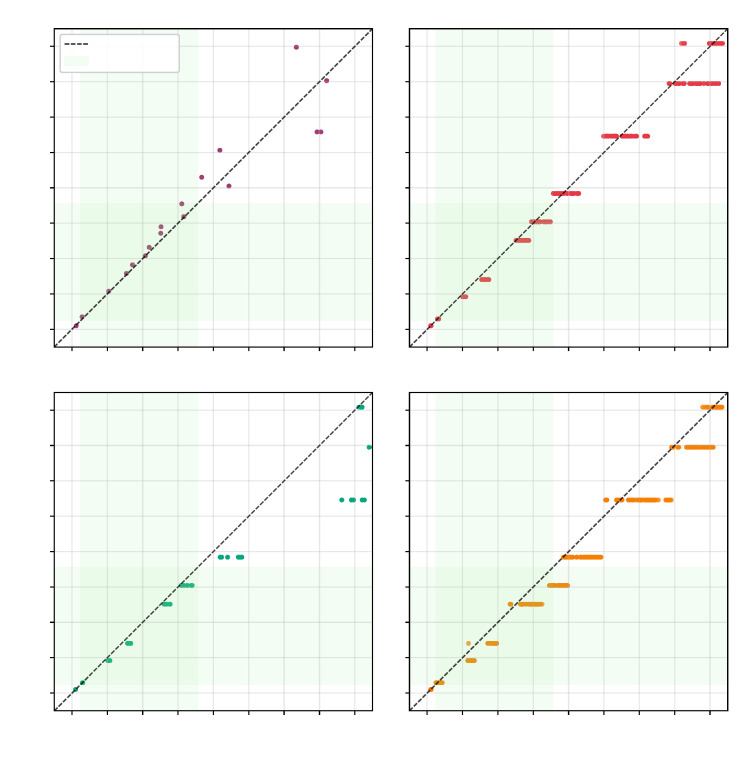
Validation dataset: predicted permittivity (1 LCSMS unit: validation, model: Polynomial D2; NSV-based) vs. reference permittivity at 25 ± 1 °C and measured permittivities from benchmark sensors (5 units each) vs reference permittivity at 17 ± 1 °C.

**Figure 16 sensors-25-06907-f016:**
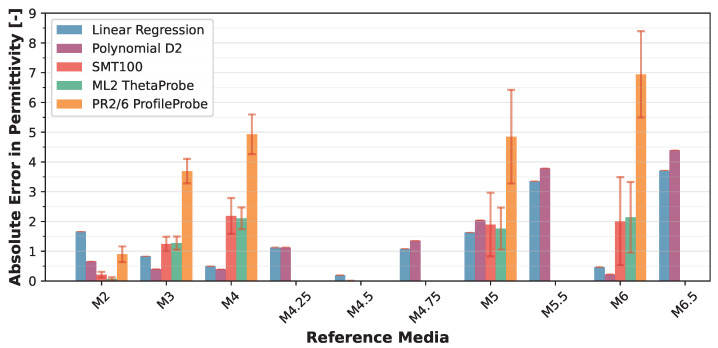
Validation dataset: absolute error (±standard deviation) in predicted permittivity (1 LCSMS unit: validation, model: Polynomial D2; NSV-based) by reference solution (at 25 ± 1 °C) compared with absolute error in predicted permittivity from manufacturer-calibrated benchmark sensors (5 units each).

**Figure 17 sensors-25-06907-f017:**
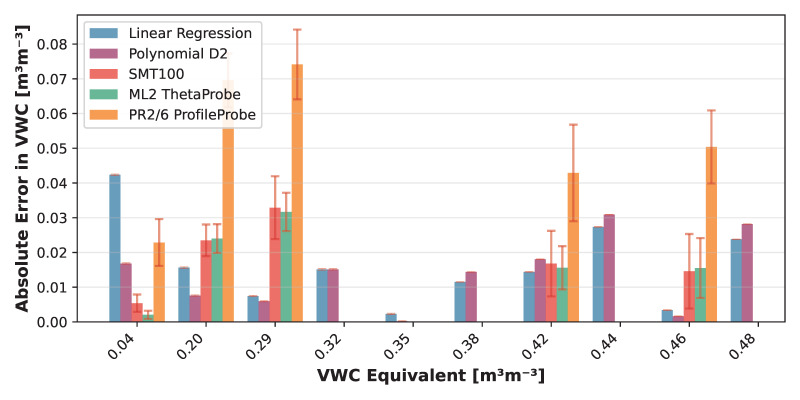
Validation dataset: absolute error (±standard deviation) in VWC inferred via Topp model (1 LCSMS unit: validation, model: Polynomial D2; NSV-based) by reference solution expressed as volumetric water content equivalent compared with absolute error in VWC from manufacturer-calibrated benchmark sensors (5 units each).

**Figure 18 sensors-25-06907-f018:**
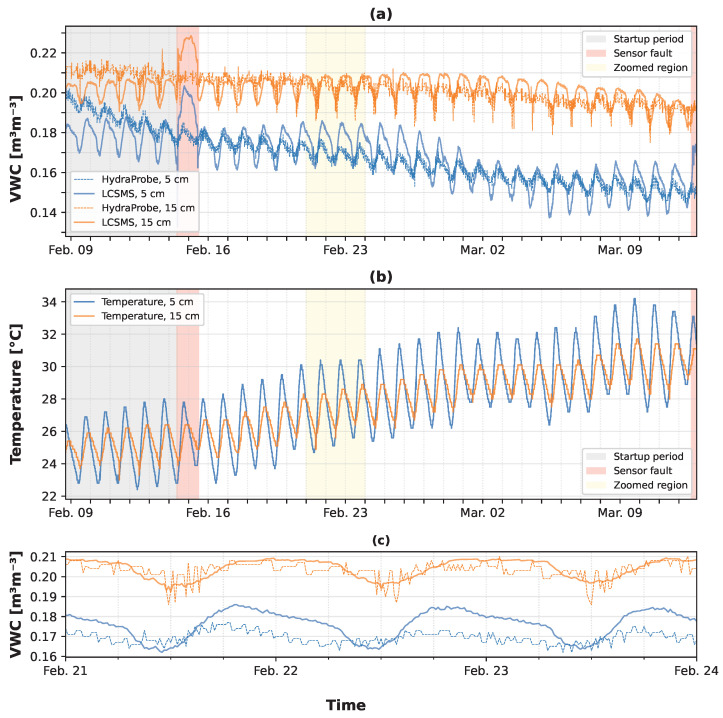
Field data from the deployment at the Berembadi site: (**a**) predicted (NSV-based Linear Regression model + Topp model) volumetric water content over time; (**b**) HydraProbe measured soil temperature over time; (**c**) predicted (NSV-based Linear Regression model + Topp model) volumetric water content over 3 days: 21–24 February.

**Table 1 sensors-25-06907-t001:** Composition and reference permittivities of reference media.

Medium	Volumetric Composition	$\epsilon_{r}$ at 25 ± 1 °C	$\epsilon_{r}$ at 17 ± 1 °C
M1	Air	1.00 ± —	1.00 ± —
M2	Glass beads	3.53 ± 0.03	2.93 ± 0.11
M3	i-C3E1	10.78 ± 0.13	9.19 ± 0.18
M4	i-C3E1:Deionised water ::0.92:0.08	15.79 ± 0.26	14.05 ± 0.24
M4.25	i-C3E1:Deionised water ::0.88:0.12	18.23 ± 0.02	—
M4.5	i-C3E1:Deionised water ::0.84:0.16	20.77 ± 0.16	—
M4.75	i-C3E1:Deionised water ::0.80:0.20	23.19 ± 0.25	—
M5	i-C3E1:Deionised water ::0.76:0.24	27.16 ± 0.15	25.13 ± 0.33
M5.5	i-C3E1:Deionised water ::0.72:0.28	28.99 ± 0.18	—
M6	i-C3E1:Deionised water ::0.68:0.32	31.84 ± 0.30	30.41 ± 0.43
M6.5	i-C3E1:Deionised water ::0.63:0.37	35.46 ± 0.38	—
M7	i-C3E1:Deionised water ::0.58:0.42	40.50 ± 0.16	38.41 ± 0.53
M7.25	i-C3E1:Deionised water ::0.53:0.47	43.00 ± 0.05	—
M7.5	i-C3E1:Deionised water ::0.43:0.57	50.63 ± 0.35	—
M8	i-C3E1:Deionised water ::0.38:0.62	55.81 ± 0.21	54.60 ± 0.46
M9	i-C3E1:Deionised water ::0.18:0.82	70.31 ± 0.17	69.51 ± 0.63
M10	Deionised water	79.77 ± 0.23	80.83 ± 0.53

**Table 2 sensors-25-06907-t002:** Summary of model validation performance metrics for VLT-based and NSV-based calibrations. Overall metrics computed across full permittivity range (1.0–79.8); Target metrics for range 2.5–35.5. All models trained with 12,075 samples (11,875 in target range) and validated on 83 unique test samples.

Model	R^2^ Overall	R^2^ Target	MAE $\epsilon_{s}$ Overall	MAE $\epsilon_{s}$ Target	MAE VWC Target
*VLT-based models*					
Linear Regression	0.911	0.961	3.99	1.37	0.02
Polynomial D2	0.909	0.954	3.91	1.37	0.02
*NSV-based models*					
Linear Regression	0.912	0.965	3.97	1.29	0.02
Polynomial D2	0.910	0.958	3.89	1.24	0.01

**Table 3 sensors-25-06907-t003:** Comparison of mean absolute errors (MAE) in predicted permittivity and volumetric water content (VWC) for manufacturer-calibrated and air–water normalisation-calibrated benchmark sensors.

Sensor	MAE $\epsilon_{s}$ Overall	MAE $\epsilon_{s}$ Target	MAE VWC Target
*With manufacturer’s calibration*			
SMT100	2.13	1.51	0.02
ML2 ThetaProbe	11.88	3.84	0.04
PR2/6 ProfileProbe	9.25	2.77	0.03
*With air–water normalisation calibration*			
SMT100	1.81	1.23	0.02
ML2 ThetaProbe	5.59	1.47	0.02
PR2/6 ProfileProbe	4.20	4.26	0.05

## Data Availability

All the data, including the code to produce the plots presented, are archived to the Zenodo repository: https://doi.org/10.5281/zenodo.17436153. The open-source code for setting up the picoSMMS is hosted at the Zenodo repository: https://doi.org/10.5281/zenodo.17436381, but it can also be found in the GitHub repository: https://github.com/veethahavya-CU-cz/picoSMMS/releases/tag/publication_v2.0.

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
