# Peer review of "picoSMMS: Development and Validation of a Low-Cost and Open-Source Soil Moisture Monitoring Station"

_sensors, 2025, doi:10.3390/s25226907_

Round 1
Reviewer 1 Report
Comments and Suggestions for Authors
in attached file

Author Response
Kindly see the file attached for our responses to the comments of Reviewer 1.

Reviewer 2 Report
Comments and Suggestions for Authors
This manuscript focuses on the challenges posed by the strong spatiotemporal variability of soil moisture and the cost constraints that hinder high-density monitoring. It develops a fully open-source, low-cost, off-grid soil moisture monitoring station, picoSMMS, and implements a robust, temperature-aware, sensor-independent calibration of a commonly used low-cost capacitive sensor. When converted to volumetric water content (VWC) using the Topp equation, the error is relatively small, comparable to that of mid-range commercial sensors, demonstrating cost-effectiveness for dense deployment. However, the validation period was short, and further long-term testing and evaluation of salinity effects are still needed. Overall, the reviewers believe the manuscript could be published in Sensors after major revision. Specific comments are provided below:
- The keywords should be expressed in a more professional and specific manner; for example, the overly general term “data logger” should be replaced with a more precise one.
- The authors’ justification for discarding higher-order polynomial/tree/SVR models as “overfitting or causing step effects” is insufficient. It is recommended to add a model comparison table (including parameter counts, cross-validation scheme, tuning ranges, final scores, and residual characteristics).
- In the title “picoSMMS: Development and Validation of a Low-Cost and Open-Source Soil Moisture Monitoring Station,” the capitalization of “picoSMMS” should be standardized according to the official naming convention (e.g., initial capital letter or consistent with usage in the text).
- Please clarify the physical basis and applicable range of the linear temperature term, provide point estimates and confidence intervals for the temperature coefficient, and include a temperature–error sensitivity curve.
- In Figure 13, the VWC units mix m³/m³ and “%”, and the y-axis label “Absolute Error in VWC [m³ m⁻³] (%)” appears inconsistent. It is recommended to standardize throughout the manuscript (including Fig. 18) by using m³/m³ as the primary unit, with “%” given in parentheses.
- The authors are advised to further explain how probe sensitive volume, installation depth, and contact conditions were handled for equivalence, to ensure comparability between probes and setups.
- Since field records may be affected by voltage spikes and interruptions, it is recommended to extend the validation period to the seasonal scale, in order to assess long-term stability and robustness.
- It is suggested to improve the readability of Figures 10–15 by increasing font and line width, and enlarging shading/legend elements.
- Although the abstract reports errors and comparisons, it lacks specification of the “usage conditions.” It is recommended to add details of the testing range (dielectric, temperature, soil types) and explicitly note limitations, to avoid over-generalization by readers.
- The references in this paper are relatively few and do not adequately reflect the latest progress and developments in related fields at home and abroad. It is suggested to add the following references to reflect the timeliness and depth of the paper.
https://doi.org/10.1016/J.CATENA.2025.109206
https://doi.org/10.3390/S24248156
https://doi.org/10.3390/S24185958
https://doi.org/10.3390/S24185886
Author Response
Kindly see the file attached for our responses to the comments of Reviewer 2.

Reviewer 3 Report
Comments and Suggestions for Authors
This manuscript develops picoSMMS, an open-source soil moisture monitoring station using low-cost capacitive sensors. The calibration approach relates sensor output to bulk dielectric permittivity using reference fluids. While the work addresses affordable soil moisture monitoring, several issues require attention.
1. There is insufficient field validation. Deployment lasted ~1 month before system failure. Both Version 1 stations failed (Rokytka ~3 weeks, Berembadi ~4 weeks) due to voltage spikes. Version 2 improvements are presented, but no field testing data.
2. The salinity investigation seems flawed. Section 3.1 tests only at full saturation (VWC=1, epsilon~80) where uncertainty is highest. This doesn't support claims of "negligible effect."
3. The inter-sensor variability assessment isn't sufficient. Only two LCSMS sensors tested (one calibration, one validation). Given cited studies report significant unit-to-unit variability, this is insufficient to validate the normalization approach.
4. Temperature protocol inconsistency. Benchmark sensors tested at 17 +/-1 C (Fig. 14), LCSMS at 25 +/-1 C (Figs. 10-11). Normalization applied to commercial sensors is relegated to Appendix E.3 rather than properly discussed.
5. Missing QC documentation. No details on outlier detection, data exclusion criteria, temperature correction procedures, or handling of voltage spike readings.
6. The Abstract claims accuracy "acceptable for many scientific applications" but doesn't define which applications or acknowledging limitations.
7. The statistical analysis is weak, with no formal testing of sensor differences or model comparisons. Linear vs. Polynomial D2 choice based on visual inspection, not rigorous or principled criteria used.
8. Reproducibility seems to be incomplete, despite emphasis on open-source principles, component sourcing not fully documented, 3D printing parameters missing, assembly procedures cursory, IPython workflow not clearly summarized.
9. The introduction focuses heavily on LCSMS studies but doesn't adequately covers broader open-source monitoring systems and recent calibration methodology work.
10. The field implementation isn't clear. Temperature correction approach for field applications is not explained, especially the spatial relationship between temperature and moisture sensors.
11. There are some issues with VWC conversion as it uses generic Topp equation without adequately considering additional errors this introduces.
12. The "sensor-independent" claim is misleading and calibration specific to DFRobot SEN0193. Normalization still requires individual sensor measurements in air and water (sensor-unit-specific, not sensor-independent).
The laboratory calibration work seems reasonable, but the manuscript requires substantial revision to address hardware reliability, field validation, and methodological concerns before acceptance.
Author Response
Kindly see the file attached for our responses to the comments of Reviewer 3.

Round 2
Reviewer 1 Report
Comments and Suggestions for Authors
please found my review attached

Author Response
Kindly see the attached file for our responses.

Reviewer 3 Report
Comments and Suggestions for Authors
The authors have addressed the majority of my concerns by either updating the text or justifying the absence of a specific detail from the text.
The authors claim to have replaced "sensor-independent" with "sensor-unit-specific" throughout the manuscript. However, the abstract still uses the confusing phrase "sensor-unit-independent calibration" which combines both terms seemingly contradictorily.
The fundamental limitation of inter-sensor variability (only two sensors tested) remains a significant weakness that limits the practical applicability of the calibration approach, though the authors now acknowledge this transparently.
Author Response

(The authors gave the same response as above.)
